

# Laminar gas inlet – Part 2: Wind tunnel chemical transmission measurement and modelling

Da Yang[1,2,3,*], Emmanuel Assaf[4], Roy Mauldin[4], Suresh Dhaniyala[3] , and Rainer Volkamer[1,2,4,*]

[1]Department of Chemistry, University of Colorado Boulder, Boulder, CO
[2]Cooperative Institute for Research in Environmental Sciences (CIRES), University of Colorado Boulder, Boulder, CO
[3]Mechanical and Aeronautical Engineering, Clarkson University, Potsdam, NY
[4]Dept of Atmospheric and Oceanic Sciences, University of Colorado Boulder, Boulder, CO

*Correspondence to: Da Yang (da.yang@colorado.edu); Rainer Volkamer (rainer.volkamer@colorado.edu)

**Abstract.** Aircraft-based measurements of gas-phase species and aerosols provide crucial knowledge about the composition and vertical structure of the atmosphere, enhancing the study of atmospheric physics and chemistry. Unlike aircraft-based aerosol particle sampling systems, the gas loss mechanisms and transmission efficiency of aircraft-based gas sampling systems are rarely discussed. In particular, the gas transmission of condensable vapors through these sampling systems requires

systematic study to clarify the key factors of gas loss and to predict and improve gas sampling efficiency quantitatively. An aircraft gas inlet for aircraft-based laminar sampling of condensable vapors is described in part 1 (Yang et al., 2024), which describes the inlet dimensions, flow analysis and modelling, along with initial gas transmission estimates. Here we test and characterize the complete inflight sampling system using for gas-phase measurements of $H_2SO_4$ in a high-speed wind tunnel, and conduct detailed computer fluid dynamics (CFD) simulations to assess inlet performance under a range of flight conditions.

The gas transmission efficiency of $H_2SO_4$ through different sampling lines was measured using Chemical Ionization Mass Spectrometry (CIMS), and the experimental results are reproduced by the CFD simulations of flow and mass diffusion using a mass accommodation coefficient, $\alpha_i = 0.70 \pm 0.05$ for $H_2SO_4$ on inlet lines. The experimental data and simulation results show consistently that gas transmission efficiency increases with an increased sampling flow rate. The simulation results further indicate that sampling efficiency can continue to improve to a certain level after the sampling flow enters the turbulent

flow regime, up to Reynolds numbers, Re ~ 6000. A decrease in transmission is predicted only for higher Re numbers. These results challenge the widely held assumption that laminar flow core sampling is the best strategy for sampling condensable vapors. The gas-phase $H_2SO_4$ transmission efficiency can be optimized (increased by a factor ~2) by minimizing residence time, rather than maintaining laminar flow; this benefit extends to other condensable vapors and applies over the full range of operating conditions of the aircraft inlet system. For a sticky species ($\alpha_i > 0.25$), the laminar diffusivity is important to predict

the transmission efficiency via the aircraft inlet section, while for less sticky species ($\alpha_i < 0.25$) the gas-phase diffusivity plays a minor role in predicting the gas transmission efficiency in the sampling line.



## 1 Introduction

Knowledge of the composition of the atmosphere and its change over time is relevant to public health, air quality and climate. Atmospheric composition research requires measurements and understanding of both its gas-phase and aerosol constituents. A particular analytical challenge is the in-situ aircraft-based measurement of condensable vapors, which critically contribute to both aerosol formation through nucleation and its early growth processes. During sampling, condensable vapors such as sulfuric acid can be efficiently lost to pre-existing aerosol surfaces and the inner walls of inlet lines. Such losses alter and distort downstream measurements of the properties of sampled gas and aerosol species. The accurate sampling of vapors that form aerosols is relevant because they ultimately affect our measurement of ambient aerosol properties and thus human health impact, as airborne particle impact on human health is size and composition-dependent, . In the context of air quality, aerosol-forming vapors hold potential to contribute to haze and reduce visibility, posing risks to both urban and rural environments. Additionally, aerosols influence climate by scattering and absorbing sunlight, and by acting as cloud condensation nuclei, which can alter cloud properties and precipitation patterns. Furthermore, trace gases are relevant for atmospheric chemistry as they impact the formation and depletion of ozone, oxidative capacity, and the oxidation of mercury, a potent neurotoxin (Khalizov et al., 2020; Shah et al., 2021). Understanding the formation and growth of short-lived reactive gases, and condensable vapors that can form aerosols is thus essential for addressing challenges in public health, air quality, and climate change.

Aircraft-based measurements are necessary and valuable for measuring atmospheric composition directly, and provide the vertical structure of constituents from the surface into the upper troposphere and lower stratosphere (UTLS) with a high temporal and spatial resolution (Brenninkmeijer et al., 1999; Filges et al., 2015; Karion et al., 2010). These measurements complement and inform remote sensing observations on global scales, provide process level insights into atmospheric chemistry and physics, and are useful to assess predictions from chemical transport models about air quality and climate.

The aircraft gas inlet is one of the key components of an aircraft-based sampling system. It serves as the interface between the ambient atmosphere and the analytical instruments aboard the aircraft, facilitating the collection of representative samples for analysis. Despite the advanced knowledge of measurement instruments (Kulkarni et al. 2011, Clemitshaw 2004), particle sampling methodologies (Kulkarni et al., 2011; Von der Weiden et al., 2009; Yang, 2017) and particle sampling inlets (Craig et al., 2014; Dhaniyala et al., 2003; Eddy et al., 2006; Moharreri et al., 2014) , the studies of gas sampling inlet (Fahey et al., 1989; Kondo et al., 1997; Ryerson et al., 1999; Yang et al., 2024) and gas transportation process are limited. More specifically, there is a lack of understanding of the gas sampling loss of condensable vapors. The gas sampling loss can occur due to the inlet design, the properties of gas-phase species, and the operating conditions. Loss of condensable vapors can result in uncertainties in the measured concentrations, thereby reducing the accuracy and reliability of atmospheric data obtained from airborne platforms.




Understanding and quantifying gas sampling efficiency is essential for improving the accuracy of atmospheric measurements and enhancing our understanding of atmospheric processes. As the gas sample moves through the sample tube, gas-phase diffusion will drive gas-species in the sampled flow from a high concentration region to a low concentration region (Bird et al., 2006). For studying gas-phase diffusion in complex geometries, computational fluid dynamics (CFD) simulations have

been used, and their performance validated in pipe-like geometries (De Schepper et al. 2008; 2009; Deendarlianto et al. 2016; López et al. 2016). Applications of CFD method on modeling diffusion in gas-liquid, gas-solid and other two-phase flow regions are well applied and discussed (Hassanzadeh et al. 2009; Xin et al. 2015). Fick's law as a fundamental model of diffusion in mass transport is well studied for binary diffusion (Bird et al. 2006). As studies of diffusion are relevant to many different fields, Fick's law has been revised and adjusted for different applications (Lowney and Larrabee 1980; Van De Steene

and Verplancke 2006). For our study, as demonstrated in part 1 of our inlet study (Yang et al., 2024), the prediction of turbulent diffusivity has a significant impact on the mass transport model applied in the aircraft inlet. Yang et al. (2024) establishes that using the correct flow model is a critical pre-requisite for accurately predicting gas loss. Consistent with our previous study of modelling water vapor transport in the aircraft inlet, the same mass transport model is used to parametrically characterize gas phase $H_2SO_4$ loss in our sampling system.


In this paper, we aim to investigate the mass diffusion loss associated with our aircraft-based gas sampling inlet and sampling line. Using a combination of CFD simulations and gas-phase $H_2SO_4$ measurements in a high-speed wind tunnel, we seek to elucidate the relationship between gas sampling loss and sampling conditions, and ultimately provide the strategies to optimize the gas sampling efficiency through our aircraft-based sampling system. Section 2 describes our methodologies to study the

gas-phase $H_2SO_4$ transmission through our sampling system. This includes both the experimental setup to measure gas-phase $H_2SO_4$ transmission in the wind tunnel, and the procedure of computational simulations. Section 3 presents and discusses our experimental and CFD model results, the comparisons to evaluate the simulation data using the experiments, and the predictions of gas-phase $H_2SO_4$ sampling efficiency through our sampling system. Finally, section 4 summarizes our findings and presents conclusions as well as an outlook.

**2 Method**

The complete aircraft gas-sampling system includes: the aircraft gas inlet described as part of Yang et al. (2024), the aircraft sampling line, and a $NO_3^-$ API-LToF-CIMS (Atmospheric Pressure Interface Long Time of Flight Chemical Ionization Mass Spectrometer with nitrate reagent ion) instrument. The nitrate ToF-CIMS was successfully utilized to measure gas-phase sulfuric acid ($H_2SO_4$), hydroxyl radicals (OH), and various other acids (Eisele and Tanner, 1991, 1993; Tanner et al., 1997).

This technique has proven effective in providing accurate and sensitive measurements of these critical atmospheric components. Our inlet design is based on the design by Eisele et al. (1997). It incorporates two shrouds to isolate and straighten



the sample flow, with the sampling tube positioned at the center of the inner shroud to collect the sample flow without wall losses from the shrouds. The back of the inlet features a restrictor to slow down and control the flow rate. This design can facilitate in situ calibration of $OH$, $H_2SO_4$, and other species, such as iodic acid (Eisele et al., 1997; Finkenzeller et al., 2023; Mauldin et al., 1998). Gas-phase $H_2SO_4$ is measured through the sampling system under different experimental configurations of the inlet (sampling line, restrictor size); and a range of different wind conditions and sampling flow rates were tested. The CFD flow and mass transport simulations represent the sampling system, boundary conditions and sampling flow rates. Both experimental results and simulation results were further analyzed to derive and compare the gas transmission efficiency.

**2.1 Wind tunnel experimental setup**

The sampling system of gas-phase $H_2SO_4$ measurements is shown in Figure 1. The inlet is installed inside a wind tunnel test section that is 0.3m by 0.3m in dimensions (Fig. 1a). The wind tunnel is operated at different freestream velocities (30, 75, 130 and 180 m s$^{-1}$) that represent the range of aircraft speeds. This generates the different external and internal flows of the inlet. The flow from the sample tube was further sub-sampled to transfer $H_2SO_4$ through a sampling line to the CIMS instrument (Fig. 1b). The sampling tube shown in Figure 1b is the actual sampling line used in-flight aboard the NSF/NCAR GV aircraft; it consists of a 33" (0.84 m) long line with several bending sections built to fit the requirements of installation on the aircraft. The CIMS instrument was located directly outside the wind tunnel testing section (Fig. 1c) and connected via the sampling line. The CIMS measured real-time chemical signal in response to the changes of wind tunnel conditions and sampling conditions. A mass flow controller (MFC) is used to throttle a pump to maintain and vary the sampling mass flow rate. In addition, to evaluate the role of shape of the sampling line on the gas-phase $H_2SO_4$ transmission efficiency, three other types of sampling lines are also tested. The schematic diagram of the tested lines is shown in Figure 1d. The in-flight sampling tube with bends is labeled as type 1. The sampling tube labeled type 0 is a 33" (~0.84m) straight tube of the same total length as the in-flight tube (i.e. type 1). The type 2 and type 3 tubes are the 40" (~1m) straight tubes connected at the end of 33" (~0.84m) straight tube with and without 90° bend at the end, respectively. We note that the wind tunnel setup has an extra 90 degree bend compared to actual in-flight sampling system. This extra bend is identical in all sampling lines and thus cancels out in relative comparisons of transmission between different line type configurations. Since the overall inlet transmission for the in-flight sampling system relies on piecing together the individual segments of the inlet assembly as shown in Figure 6, the extra bend is not counted towards assessments of the aircraft configuration.



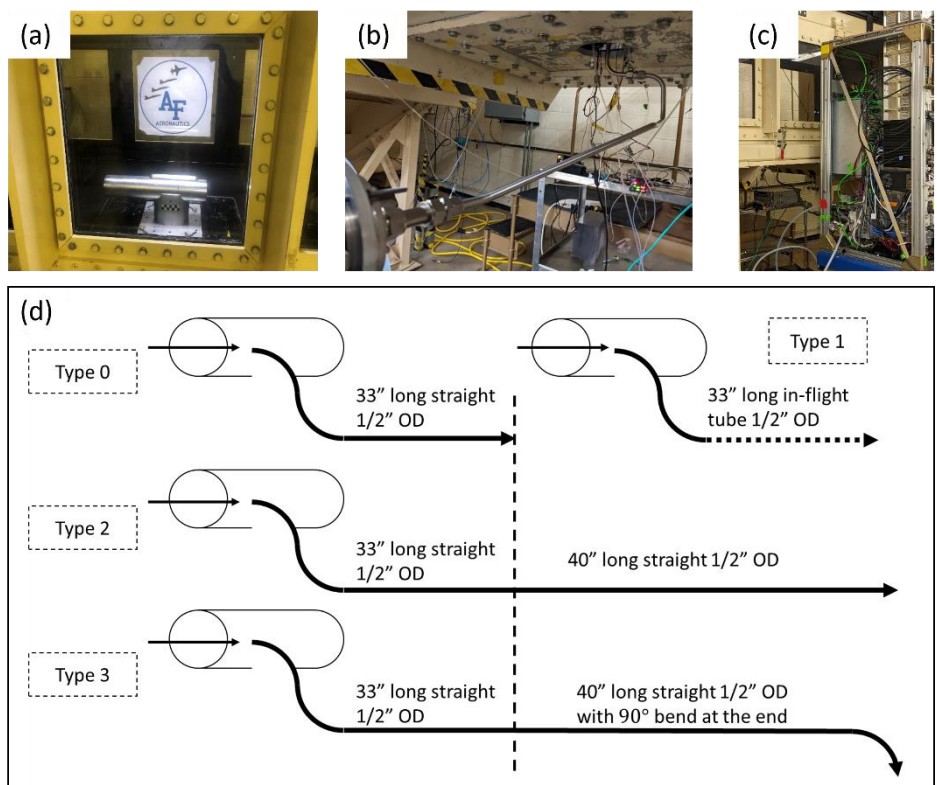

**Figure 1. The wind tunnel experimental setup for $H_2SO_4$ transmission measurements. (a) Laminar gas inlet installed inside the wind tunnel test section. (b) The sampling lines used to connect the inlet with the CIMS instrument (c). The CIMS measured $H_2SO_4$ produced from the photolysis of H₂O at the tip of sampling tube inside the inlet under different line configurations, and wind tunnel conditions. (d) The schematic of four tested types of sampling lines; the line used in-flight is shown in (b) and referred to as type 1.**

## 2.2 Wind tunnel experiments

To conduct gas-phase $H_2SO_4$ measurement, additional supporting devices are integrated into the inlet (Fig. 2). As shown in Figure 2, a UV source consisting of a Pen-Ray Mercury Lamp (90-0012-01), illuminates the inner shroud ~10mm in front of the sampling tube entrance. Ambient water vapor from the free stream is photolyzed as it travels into the inlet through the outer shroud. The photochemical dissociation (photodecomposition) of water vapor upon interaction with the UV light generates hydroxyl radicals.

$$H_2O + h\nu \rightarrow OH + H$$

The center of the inner shroud flow is sub-sampled by the sampling tube. $SO_2$ is injected via a port located near the entrance of the sampling tube. The hydroxyl radical interacts with $SO_2$, oxygen and $H_2O$ to generate $H_2SO_4$ (Finlayson-Pitts and Jr, 1999; Kolb et al., 1994). The gas-phase $H_2SO_4$ travels through the sampling system and is detected as NO₃⁻ clusters; $(HNO_3)_n \cdot NO_3^-$ where n=0, 1 ,2. To determine the amount of $H_2SO_4$, the main reactions involved $(HNO_3)_n \cdot NO_3^-$ clusters, where n=0 and 1. Therefore, the measurements of species $HSO_4^-$ and $HNO_3 \cdot HSO_4^-$ were primarily analysed.

$$NO_3^- + H_2SO_4 \rightarrow HNO_3 + HSO_4^-$$



$$HNO_3 \cdot NO_3^- + H_2SO_4 \rightarrow HNO_3 + HNO_3 \cdot HSO_4^-$$

In addition, to emulate the different operating conditions and to examine the impact from different system designs, four parameters were varied in the wind tunnel experiments, i.e., restrictor size of the aircraft inlet, free stream velocities ($U_\infty$), sampling line type, and sampling flow rate. The testing cases are summarized in table 1.

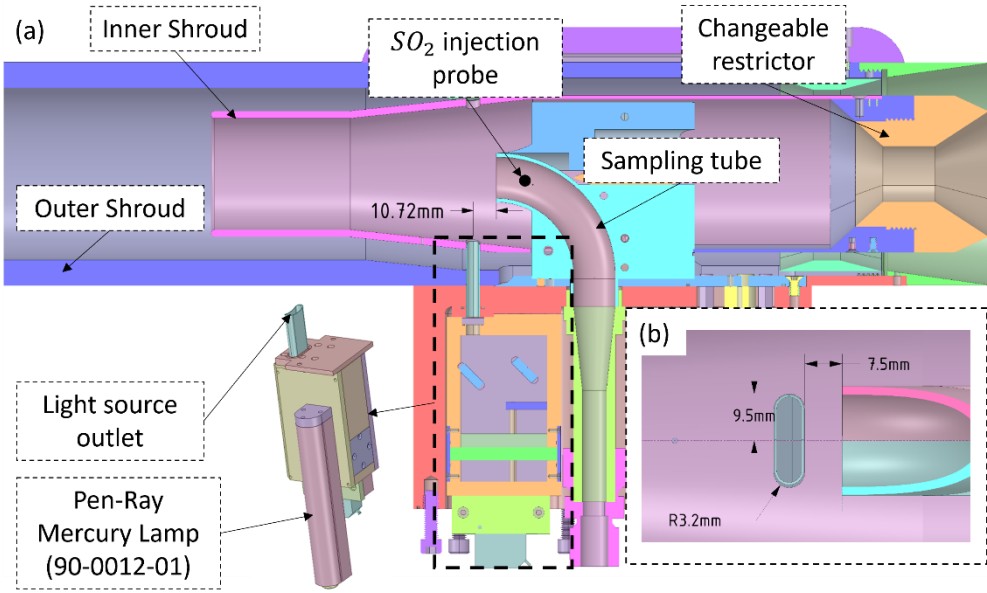


**Figure 2. Cross-section view of the laminar gas inlet with UV lamp. (a) cross-section side view: shows different components of the inlet, incl. a 3d view of the UV source with mirror housing. (b) Cross-section top view: shows a detail view of the UV source outlet placement in front of the sampling tube entrance with dimensions.**

**Table 1. Variables of the wind tunnel $H_2SO_4$ gas measurements.**


| Restrictor size (mm) | 12.5 | | | | | | | | 17 | | | | | | | | | | | |
|---|---|---|---|---|---|---|---|---|---|---|---|---|---|---|---|---|---|---|---|---|
| $U_\infty$ $(m\,s^{-1})$ | 30 | | 75 | | 130 | | 180 | | 30 | | | | 75 | | | | 130 | | 180 | |
| Sampling line type | 0 | 1 | 0 | 1 | 0 | 1 | 0 | 1 | 0 | 1 | 2 | 3 | 0 | 1 | 2 | 3 | 0 | 1 | 0 | 1 |
| Sampling flow rate (SLPM) | 4, 8, 10 | 4, 8 | 4, 8, 10 | 4, 8 | 4, 8, 10 | 4, 8 | 4, 8, 10 | 4, 8 | 4, 8, 16 | 4, 8 | 4, 8, 16 | 4, 8, 16 | 4, 8, 16 | 4, 8 | 4, 8 | 4, 8, 16 | 4, 8 | 4, 8 | 4, 8 | 4, 8 |



## 2.3 Computational flow and mass transport modelling for sampling line

To inspect the correlations between internal flow features and gas measurement results, the CFD simulation results from previous inlet studies were used. The details of the aircraft inlet CFD modelling was described in our first laminar gas inlet

paper (Yang et al., 2024). Compared to the freestream conditions considered in the first paper , since gas-phase $H_2SO_4$ measurements were also conducted at freestream velocity of 30 m s$^{-1}$;in gas measurement experiments, this freestream condition was added into our aircraft inlet simulations for both 12.5mm and 17mm restrictors. These additional simulations help facilitate direct comparisons with experimental results.

To predict and compare the gas transmission efficiency with measurements, we conduct flow and mass transport modelling in different sampling tube designs use the commercial code FLUENT 18.1 (ANSYS, NH) to in the 40" (~1 m) long straight tube with ID=10.7mm. For the flow modelling, as most sampling flow rates (Tab. 1) result in a Reynolds number less than 2300 in the tube, the laminar flow model is used. In addition, considering the case when Reynolds number is close to 2300, a small disturbance in the tube can cause turbulent diffusion loss, we also used the transition SST k-ω model for flow modelling across

both laminar and turbulent flow regimes ($400 < Re < 7000+$). Both flow models were used to examine the influence of diffusion coefficient in mass transport with and without any turbulence effect. Due to the symmetric nature of the tube, both a 2D-axisymmetric solver and a 3D solver were tested. The mesh is refined at the near wall region by applying bias mesh gradient for the 2D model and inflation method for the 3D model. The final mesh size was selected such that any further refinement resulted in a less than 1% change in the flow velocity profile and mass fraction profile at the outlet of the tube. The final CFD

model with a 3D geometry used ~1.6 million cells to simulate flow under different boundary conditions.

To model gas-phase mass transport, we consider the role of mass diffusion due to the concentration difference of gas-phase $H_2SO_4$ between the sample flow and the tube wall. In addition, we consider the diffusion process as binary diffusion, i.e., the gas-phase $H_2SO_4$ is diffusing in the dry air. The binary diffusivity of $H_2SO_4$ in the laminar flow regime, refer as $H_2SO_4$ laminar

diffusivity, is set as constant value of $1.04e^{-5}\ m^2\ s^{-1}$. The Fick's law of diffusion is well applied for binary diffusion, and the model is applied in FLUENT 18.1. In addition, as the temperature gradient in the transmission line is insignificant, we neglect thermal diffusion loss. The simplified governing equation of our mass transport model expressed as:

$$\frac{\partial \rho \vec{w_i}}{\partial t} + \nabla \cdot (\rho \vec{v} \vec{w_i}) = -\nabla \cdot \vec{j_i}$$

where the mass fraction $w_i$ represents the relative density of the diffusive species $\rho_i$ $(kg\ m^{-3})$ to the mixture density

$\rho$ $(kg\ m^{-3})$, i.e. $w_i = \frac{\rho_i}{\rho}$. The $v$ $(m\ s^{-1})$ represents the flow velocity. The expressions of mass flux $j_i$ is determined by different flow models.

$$j_i = -\rho D_{ij} \nabla \vec{w_i}\ ; laminar\ flow\ model$$

$$j_i = -(\rho D_{ij} + \rho D_t) \nabla \vec{w_i}\ , turbulent\ flow\ model$$





where $D_{ij}$ ($m^2\ s^{-1}$) represents the laminar diffusivity, the subscript $i$ represents the studied species (gas-phase $H_2SO_4$ or water vapor), the subscript $j$ in our study represents dry air, and $D_t$ ($m^2\ s^{-1}$) represents the turbulent diffusivity.

For the laminar flow model, the laminar diffusivity $D_{ij}$ is a constant value and only determined by the diffusing species, here gas-phase $H_2SO_4$. For the turbulent flow model, both laminar and turbulent diffusion contribute to the mass flux. The turbulent diffusivity $D_t$ is calculated based on the turbulent viscosity predicted from the turbulent flow model.

The different sampling conditions are modelled to investigate the factors that influence the gas transmission efficiency. The sampling flow rate and the mass fraction of $H_2SO_4$ on the wall are two key factors we investigated. And different mass accommodation coefficients, $\alpha_i$, were used to represent different boundary conditions on the wall for mass transport modelling. The entire set of boundary conditions for flow modeling are shown in Table 2. The mass accommodation coefficient is defined as number of gas molecules taken up by the surface divided by number of gas-surface collisions (Finlayson-Pitts and Jr, 1999). The varying boundary conditions of mass fraction of $H_2SO_4$ on the wall relates to varying α values. For our mass transport modeling, α represents the mass accommodation coefficient of a studied species diffused in the dry air. When $\alpha_i = 1$, the mass fraction of $H_2SO_4$ set on the wall is 0, the tube wall is a perfect sink and provide us the worst sampling case. When $\alpha_i$ decreases, the mass fraction of $H_2SO_4$ set on the wall increase. When $\alpha_i = 0$, the mass fraction of $H_2SO_4$ on the wall has the same mass fraction of $H_2SO_4$ in the flow, and no species will be lost to the walls due to the absence of a mass concentration gradient. Moreover, the ambient conditions in the sampling line are obtained from aircraft inlet simulations under both ground level and high altitude. All simulations were modelled assuming steady state flow.

**Table 2. Boundary conditions for flow modelling in 40" straight sampling line.**

| Flow model | Laminar flow model | SST turbulent model |
|---|---|---|
| Sampling flow velocity ($m\ s^{-1}$) | 0.6, 1.2, 2.4, 3.6 | 0.6, 1.2, 2.4, 3.6, 4.8, 6, 7.2, 8.4, 9.6, 10.8, 12, 13.2, 14.4 |
| Mass accommodation coefficient ($\alpha_i$) | 1, 0.75, 0.65 | 1, 0.75, 0.65, 0.5, 0.25, 0 |

**2.4. Calculation of uncertainties**

Uncertainty analysis is crucial for experimental data. It quantifies the confidence in the results, identifies potential sources of error, and ensures the reliability and reproducibility of the findings. Here, we summarize our uncertainty analysis and the propagation method for these errors. We used the standard uncertainty to describe the experimental errors in the repeated measurements ($e$) as:



$$e = \frac{s}{\sqrt{n}}$$

where s is the standard deviation and n is the number of data points.

To propagate the error for a linear relationship, we follow

$$e_{(x_1 \pm x_2)} = \sqrt{e_{x_1}^2 + e_{x2}^2}$$

where $x_1$, $x_2$ represents two different variables, $e_{x_1}$, $e_{x_2}$ represents the standard uncertainty from each variable.

To propagate the error for a non-linear relationship of $\frac{x_1}{x_2}$, percent relative uncertainty is used.

$$\%e = \frac{e}{mean} \times 100$$

where $\%e$ represents the percent relative uncertainty which is calculated by using standard uncertainty divided the mean

value times 100. The propagation of error follows


$$\%e_{(\frac{x_1}{x_2})} = \sqrt{\%e_{x_1}^2 + \%e_{x2}^2}$$

Additionally, converting the percent relative uncertainty to uncertainty for displaying the error bar as

$$e_{\frac{x_1}{x_2}} = \frac{\%e_{(\frac{x_1}{x_2})}}{100} \times mean(\frac{x_1}{x_2})$$

## 3. Results and discussions

The CIMS measurement data was systematically analysed with different experimental setups, operating conditions, and time
periods. The CFD simulation results are used to compare and explain the observations from experimental results. The following
sections will illustrate the data process and our findings.

### 3.1 Wind tunnel measurements

The transport efficiency measurements were made by analyzing the transport of species $HSO_4^-$ and $HNO_3 \cdot HSO_4^-$ through the
different sampling line types. These ion concentrations were recorded under different operating conditions by CIMS
instrument. Although applying the calibration factor of $H_2SO_4$ can covert these relative measurements to concentrations, the
comparison between the sampling line types only relies on the use of relative information, and thus cancels out the calibration
factor for sulfuric acid without adding uncertainty of calibration factor. As Figure 3a is shown, the wind tunnel maintains the
freestream velocity at different speeds for a measurement period and thus emulates different flight conditions for the gas-phase
$H_2SO_4$ measurement. The temperature of the wind tunnel is increasing the longer the tunnel operates. Increases in temperature
can provide slight variations in relative humidity, while the absolute humidity inside the wind tunnel is set by the outside air
which re-circulates and cools the wind tunnel. The measurement of gas-phase sulfuric acid $H_2SO_4$ is presented as the sum of
$HSO_4^-$ and $HNO_3 \cdot HSO_4^-$ divided by the sum of nitrate parent ions ($H_2SO_4$ NCPS,).



$$\frac{H_2SO_4}{\sum_{n=0}^{2}(HNO_3)_n \cdot NO_3^-} \cong \frac{HSO_4^- + HNO_3 \cdot HSO_4^-}{\sum_{n=0}^{2}(HNO_3)_n \cdot NO_3^-}$$

This ion ratio signal is unitless, and referred to as normalized counts (NCPS) or reagent normalized signal. A typical calibration

factor for sulfuric acid is $5 \times 10^9$ molec cm$^{-3}$ for the source configuration used (Eisele and Tanner, 1993; Mauldin et al., 1998); NCPS multiplied by the calibration factor yields a sulfuric acid concentration. In practice, the amount of sheath and total sampling flow were adjusted to give good NCPS signal, and the calibration factor is a function of the sample flow rate. However, the exact value of the calibration factor, and its uncertainty, does not affect the results, since our further analysis only relies on the ratio of NCPS values under comparable sample flow rates. The value of the calibration factor hence cancels

out in relative statements about inlet transmission. For each measurement period, the CIMS instrument repeated three measurements of the raw data of $H_2SO_4$ NCPS with both background mode and signal mode (Fig. 3b). The background mode measures gas-phase $H_2SO_4$ background signal that is generated from dark sources, without addition of UV light. This mode provides the baseline of $H_2SO_4$ measurement under each operating condition. The signal mode provides the gas-phase $H_2SO_4$ measurement with additional hydroxyl radical, generated by photochemical decomposition of water vapor when sample air

passes the UV light source. The signal depends on the operating conditions; i.e., NCPS of sulfuric acid decreases as the free stream velocity is increased.

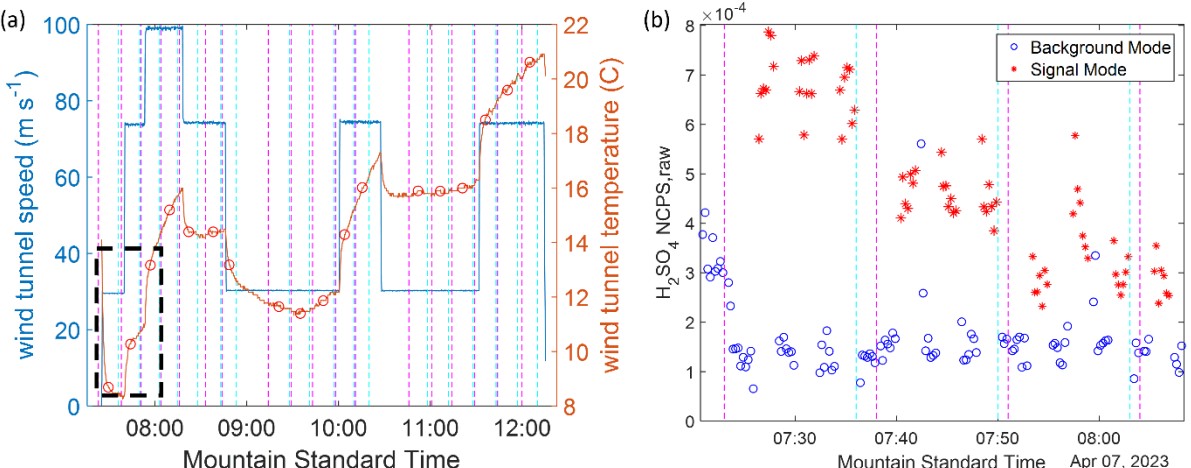

**Figure 3. Wind tunnel conditions for experiments on Apr 07, 2023. (a) (blue line) free stream velocity inside the wind tunnel; (red line) temperature inside the wind tunnel. (black circles) averages for the gas measurement periods. (b) Real time $H_2SO_4$ NCPS raw**
**measurements for the first three measurement periods marked in (a). (blue circles) background mode: detect hydroxyl radical pre-existing in the sample air; (red stars) signal mode: detect hydroxyl radical come from both sample air and water vapor photochemical decomposition. The magenta and cyan dashed lines for both figures mark the beginning and the end of each measurement case, respectively.**

The difference between signal mode and background mode is proportional to the gas-phase $H_2SO_4$ concentration that is

generated by the water vapor photolysis in the sample air ($H_2SO_4$ NCPS). Therefore, the measurement result is naturally dependent on the water vapor concentration in the ambient air. As the wind tunnel experiment was completed over multiple days, the repeated measurements under the same operating condition and setup shows significant differences that are the result



of the changes in specific humidity under different weather conditions on different experiment days (Fig. S1). These differences result in varying specific humidity inside the wind tunnel (Fig. S1a), and needed to be accounted for in comparing the

measurement results of $H_2SO_4\ NCPS$ between different experiment days. A correction factor that normalizes changes in specific humidity ($f_q$) was defined as specific humidity (q) at each measurement period divided by the mean value of specific humidity among all measurements, and is found to largely reduce the differences among repeated measurements (Fig S1b). The measurements of $H_2SO_4\ NCPS$ were normalized for constant specific humidity, $H_2SO_4\ NCPS/f_q$, for further analysis and comparisons.

**3.2 Experimental data correlation with simulation flow features**

We observed that the gas-phase $H_2SO_4\ NCPS/f_q$ measurement results still exhibited some dependence on the operating conditions of the wind tunnel, i.e., a lower normalized signal was observed when the free stream flow velocity is higher. This trend is consistently observed under different experimental configurations and operating conditions, which indicates a clear correlation between sample flow features and experimental results. To further investigate the factors at play, we studied the

internal flow features of the aircraft inlet CFD simulation results. Figure 4a shows the reverse path lines from the sampling tube, and illustrates how the sample flow that travels through the UV source illuminated area enters the sampling tube. Changing the freestream flow velocity, or the size of the restrictor, results in a different residence time inside the illuminated area (effective light area). Specifically, it results in altering the size of the region of the incoming sample flow (Fig. S2a). Moreover, the area of the sample flow, represented by its radius, that intersects with the UV illuminated area is also highly

related to the velocity ratio between the average flow velocity in the effective light area and the flow velocity at the entrance of sampling tube (Fig. S2b). The higher this velocity ratio, the larger the area of the sample flow, resulting in more hydroxyl radicals enter the sampling tube. When this velocity ratio is close to 1, i.e., near isokinetic sampling conditions are achieved at the entrance of sampling tube, the sample flow has the same radius as the radius of sampling tube entrance. In addition, our previous study (Yang et al., 2024) also observed that reaching the isokinetic sampling condition can reduce the enhancement

of turbulent intensity from incoming flow into the sample flow right inside the sampling tube entrance (Fig. S3).

To examine the correlations of normalized results ($H_2SO_4\ NCPS/f_q$) with flow features, we compared all the experimental data of the type 0 sampling line at 8 SLPM sampling flow rate with different free stream conditions. A clear correlation between the effective illuminated volume of sample flow and the measurement results ($H_2SO_4\ NCPS/f_q$) is shown in Fig. 4b. Thus,

when the upstream flow velocity is high, the sample flow volume reduces and less water molecules in the sample flow can interact with the UV source. Meanwhile, the sample flow also passes quicker through the effective illuminated area, resulting in a shorter residence time and thus less hydroxyl radicals are generated. The combination of both reasons leads to lower measured signals at high freestream flow velocity such as $180\ m\ s^{-1}$ for both sizes of restrictor. Moreover, as smaller size of restrictor causes a lower incoming flow velocity right outside the sampling tube, the experimental results ($H_2SO_4\ NCPS$) from



two restrictors shows a separate trend (left y-axis blue in Fig. 4b). To bridge the gap between the two restrictors, we calculated

the residence time of sample flow inside the effective light area from the CFD simulation results to derive the resident time

factor ($f_{RT}$). The $f_{RT}$ is calculated as the ratio of the resident time inside the illuminated area divided by the residence time of

isokinetic sampling. The further normalized measurement results ($H_2SO_4\ NCPS \times f_{RT}/f_q$) eliminates the dependence of the

signal as a function of the restrictor size. Using the above normalization, the different sampling system configurations are

compared under different operating conditions in Fig. 4b.

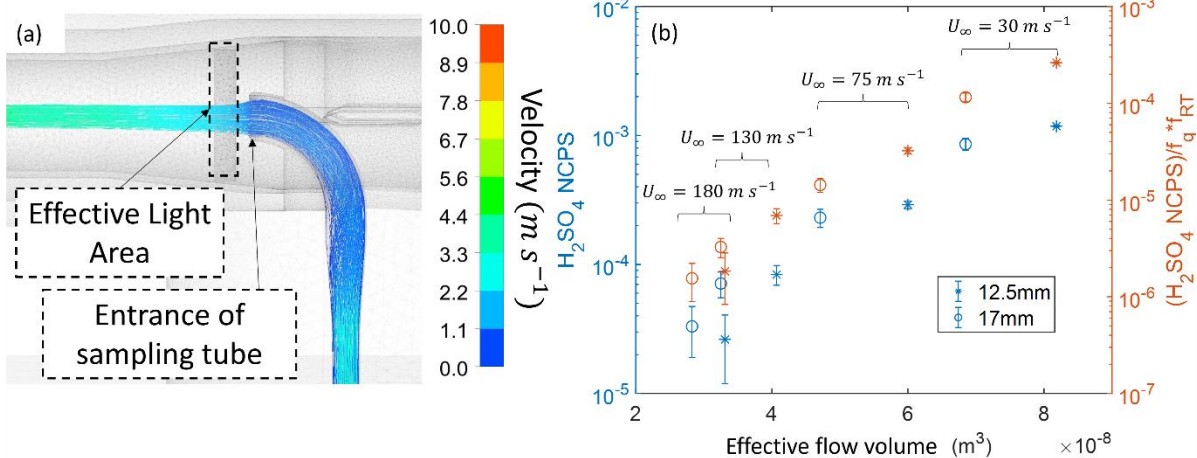

**Figure 4. Factors contribute to the quantity of measurement results. (a) The reverse flow path lines from CFD simulation at freestream flow velocity 30m/s, 12.5mm restrictor size, and 2.4 m/s sampling flow rate. The path lines are colored by the flow velocity. (b) The relationship between the measurement results of gas-phase $H_2SO_4$ with effective sample flow volume pass through light**
**area. The left y axis is $H_2SO_4\ NCPS$, the right y axis is the results from left y axis normalized with $f_q$ and multiply by the $f_{RT}$ of sample flow passing the light area. The error bar is data uncertainty which follows the description in chapter 2.4.**

### 3.3. Gas transmission efficiency comparisons

All types of sampling lines (Fig. 1d) and the different sampling flow rates are tested for gas-phase $H_2SO_4$ measurements using

the 17mm restrictor (Tab. 1). The normalized measurement results ($H_2SO_4\ NCPS/f_q$) from $30\ m\ s^{-1}$ freestream velocity is

shown in Figure 5a. The type 0 transmission line, due to the shortest length and minimum bends, shows the highest signal

under comparable sampling flow rates, compared to other types of sampling lines. Increasing the length of the sampling line

decreases the measured signal, consistent with additional losses in longer tubes. In addition, the 8 SLPM sampling flow rate

maintains the highest quantity of signal, while 16 SLPM shows the lowest signal among three different sampling flow rates.

This is related to the species reaction time and flow characteristics inside the ion molecule reaction chamber of the CIMS

instrument, and further analysis is outside the scope of this paper, which compares only ratios of normalized signal at

comparable sampling flow rates. Similar trends are also observed in all cases of $75\ m\ s^{-1}$ freestream velocity. We are choosing

the cases from the $30\ m\ s^{-1}$ freestream velocity for further analysis of the gas transmission efficiency in the sampling line due

to the higher signal-to-noise. It should be noted that upstream effects ($f_q\ and\ f_{RT}$) are accounted for in the normalization of



the signal, ensuring that the results for transmission are applicable to other free stream flow cases and restrictors of aircraft
inlet as well. For comparisons with the same restrictor, $f_{RT}$ cancels out.

The gas transmission efficiency is defined as the mass fraction (or concentration) of a molecule (here: $H_2SO_4$ or $H_2O$) at any
cross-section along the sampling pathway compared to a reference mass fraction. Based on the wind tunnel experiments, we
calculated the gas transmission efficiency of gas-phase $H_2SO_4$ measurement in the 40" (1 m) tube as the ratio of normalized
signal ($H_2SO_4$ $NCPS/f_q$) from the type 2 and type 3 sampling lines divided by the type 0, respectively, under each same
sampling flow rate ($f_{RT}$ cancels out). We use the mass fraction of the simulated $H_2SO_4$ at the outlet of the 40" straight tube
divided by its initial value at the tube inlet to calculate the gas sampling efficiency under different boundary conditions in the
model. The experimental and simulated transmission are compared in Figure 5b. The experimental results clearly show the
gas transmission efficiency in the 40" tube is far below 1 for all sampling flow rates, which indicates a gas loss in this tube. In
addition, the experimental results also show an increasing trend of the transmission efficiency as the sampling flow rate is
increased. Moreover, the simulation results over predict the observed loss under the assumption that the mass fraction of $H_2SO_4$
is 0 on the wall, which corresponds to a mass accommodation coefficient $\alpha_i = 1$. Both the laminar flow model and the SST
turbulent model have good agreement with the experimental results for accommodation coefficients between 0.65 and 0.75 for
gas-phase $H_2SO_4$. This agreement not only provides model verifications of the conclusion that reducing the residence time
during the gas-phase species transport can reduce the gas transmission loss, but also indicates the accommodation coefficient
of $H_2SO_4$ is around 65% to 75% which agrees with other studies (Hanson, 2005; Pöschl et al., 1998). In Pöschl's paper, the
mass accommodation for $H_2SO_4$ at 303K under dry conditions (RH≤ 3%) has a best fit value of 0.65, with a physical upper
limit of 1, and a lower statistical limit of 0.43. Hanson et al. further discussed the impact of RH on the mass accommodation
coefficient, and their data for $H_2SO_4$ show an average value of 0.76 for RH < 50%. Our experimental results reasonably align
closely with both previous laboratory studies, and the simulation results that apply a mass accommodation coefficient for
$H_2SO_4$ of 0.65 and 0.75 best describe the experimental observations in our aircraft inlet line. Therefore, we selected 0.7 as the
average mass accommodation coefficient and use $\pm5\%$ as uncertainty for further simulations. Moreover, the SST turbulent
model overlaps with the laminar flow model when the flow rate is low and deviates when flow is outside the laminar flow
regime. This observation provides further confidence in predicting the turbulent effects in our sampling system using the SST
model, consistent with the findings comparing hotwire measurements of turbulence described in part 1 of our inlet system
analysis (Yang et al., 2024). Overall, the flow and mass transport models provide a reasonable description of the gas
transmission loss in the entire inlet sampling system.




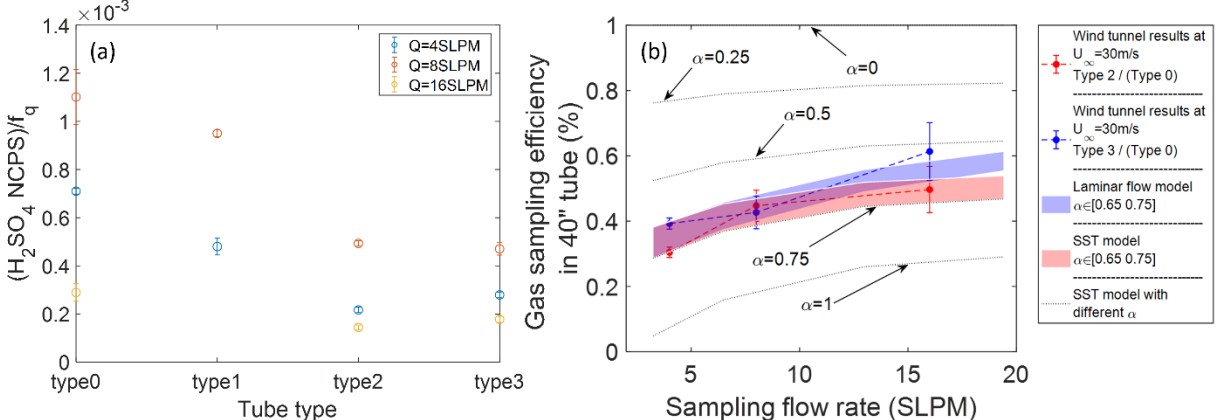

**Figure 5. Comparison of different types of sampling tubes and the gas transmission efficiency. (a) The normalized signal** ($H_2SO_4\ NCPS/f_q$) **through different types of sampling tubes are compared at a freestream velocity of 30 m/s, 17mm size of restrictor, for different sampling flow rates. (b) Comparison of the gas sampling efficiency measured along 40" (~1 m) of a straight sampling tube with simulation results that vary the accommodation coefficient, $\alpha_i$. The error bar is uncertainty of the data which follows the description of chapter 2.4.**

### 3.4 Overall gas sampling efficiency of $H_2SO_4$

The verified CFD model assumed 70% for the mass accommodation coefficient $\alpha_i$ to set the boundary condition on the surface wall for gas-phase $H_2SO_4$ transport modelling. The gas-phase $H_2SO_4$ transmission efficiency of the overall inlet sampling system is calculated as a product of the transmission in different sampling sections. Specifically, Figure 6 illustrates the individual components used to assess the overall gas-phase $H_2SO_4$ transmission efficiency: (a) transmission through the 12.5mm restrictor inlet; (b) a straight 33" (~0.84m) tube; (c) losses due to the shape factor (type 0 compared to type 1) due to the bending features (complements losses under (b)) with 10% uncertainty; and (d) a straight 20" (0.5m) tube, respectively. The overall uncertainty in the transmission of the aircraft inlet sampling system combines the uncertainty in an ideal sampling system under various flow rates and flight conditions with a $\pm 5\%$ variation of the selected mass accommodation coefficient, and the shape factor ($\pm 10\%$).

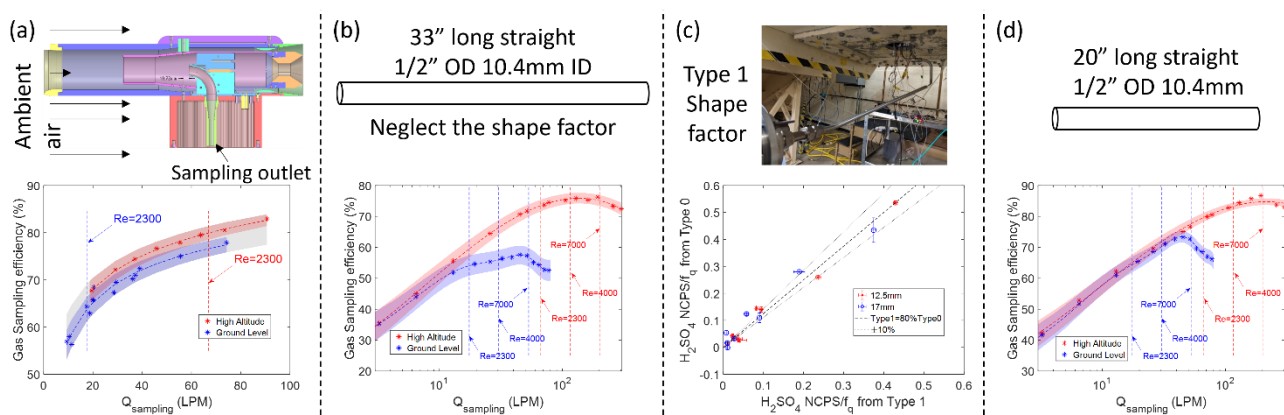





**Figure 6. The gas-phase $H_2SO_4$ sampling efficiency in each partial section of the in-flight sampling system. The shaded areas in (a), (b) and (d) presents $\pm5\%$ uncertainty of the selected mass accommodation coefficient (70%); the overall sampling efficiency is shown in Figure 7. Panel (6a) Inlet section: ambient air to the sampling outlet; (b) 33" (0.84m) straight tube; (c) Shape factor: compares type 1 and type 0 under same operating conditions; (dashed line) fitted as $Type\ 1 = 80\% \times Type\ 0$; (dotted lines) $\pm10\%$ uncertainty of shape factor; (d) 20" (0.5m) straight tube to connect the CIMS in the aircraft configuration.**

Notably, the individual sampling sections influence the gas-phase $H_2SO_4$ transmission efficiency differently. For the aircraft inlet section, the sample flow in the gas inlet is mainly impacted by the upstream turbulence. As a result, the gas sampling efficiency through this section shows a direct correlation with the sampling flow rate, and a negligible difference at the two flight altitudes investigated (Fig. 6.a). However, since the sample flow in the sampling line is directly impacted by the characteristics of the local flow, the prediction of gas transmission efficiency inside the sampling line is significantly different between ground level and high altitude (Fig. 6b & 6d). At ground level, due to the higher flow density, the flow in the sampling tube is reaching the turbulent regime at a lower volume flow rate than at high altitude. In addition, the simulation results from both altitudes consistently show that the gas sampling efficiency can increase even when the sample flow enters the turbulent flow regime ($Re > 4000$). However, when flow reaches higher turbulent ($Re > 7000$), the sampling efficiency in sampling tube starts dropping again. This observation is due to the higher prediction of turbulent diffusivity, and indicates that the fate of the overall inlet gas transmission efficiency is determined by the local sampling flow rate (Q). Since the Reynolds Number in the tube can be expressed as $Re = \frac{4Q\rho}{D\pi\mu}$, at a fixed sampling flow rate Q, the results imply that enlarging the inner diameter of the sampling tube (D) can reduce the Re and thus reduce the model prediction of turbulence, with benefits for the overall gas transmission efficiency. In order to achieve the maximum gas transmission efficiency, our simulation results suggest that the optimized flow rate should maintain the Re in the sampling tube at ~6000.

The overall gas sampling efficiency of the actual in-flight sampling system is calculated by multiplying each sampling efficiency at different sections and shape factor together. The results are shown in Figure 7. We present the overall sampling efficiency of the actual in-flight sampling system at two altitudes as a function of the sampling flow rate. The shaded areas in Figure 7 represent the largest range of uncertainty from $\pm5\%$ variation of mass accommodation coefficient and $\pm10\%$ variation of shape factor. The simulation results indicate the overall gas sampling efficiency is increasing as the sampling flow rate is increased at both altitudes, and for most sampling conditions. This effect extends even when the sampling flow is passing the laminar flow regime ($Re \leq 2300$), and reflects that the residence time inside the inlet determines overall inlet transmission well into the turbulent flow regime. However, the turbulent diffusivity predicted from the flow model increases as the Reynolds number increases. The overall sampling efficiency increases less rapidly, and eventually starts to decrease when the sampling flow rate reaches a much higher Reynolds number level ($Re > 7000$). The overall sampling efficiency is lower at ground level compared to high altitude if the results are compared at the same flow rate. This is due to sample flow is more turbulent at lower altitude. Overall, based on the simulation results, the overall sampling efficiency of gas-phase $H_2SO_4$ for the actual in-flight sampling system can increase by a factor of 2 (from 12% @ 10 SLPM sampling flow rate to ~20% @ ~30 SLPM) by





increasing the sampling flow rate at ground level. At high altitude, due to lower air density resulting into a lower Re, the overall

sampling efficiency can increase to ~40% by increasing the sampling flow rate.

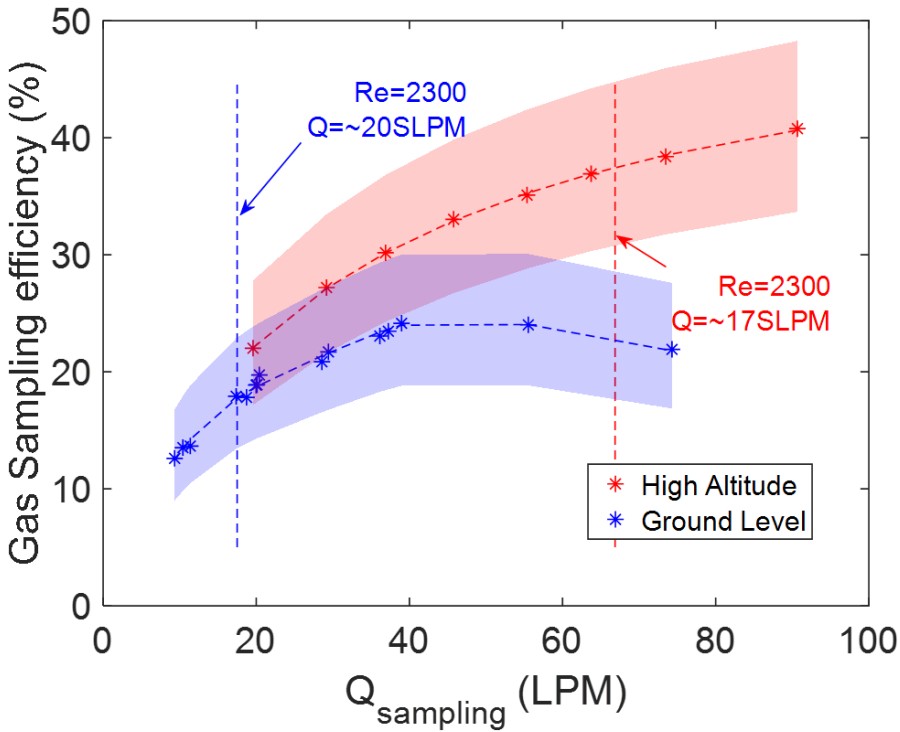

**Figure 7. The correlation between overall gas transmission efficiency of $H_2SO_4$ and sampling flow rate $Q_{sampling}$ (LPM) at ground
level and high altitude.  The dash lines mark the Reynolds Number in sampling line equal to 2300, which distinguish the laminar
flow regime and beyond. The shade areas present the range of uncertainty from $\pm5\%$ variation of mass accommodation coefficient
and $\pm10\%$ variation of shape factor.**

**3.5 Extending transmission to other species**

The transmission efficiency of $H_2SO_4$ can be extended also to other species. However, a detailed treatment of other species
requires a careful consideration of diffusivity and mass accommodation coefficients that is beyond the scope of this work. We

illustrate the effects using water vapor as an example molecule with higher diffusivity, and discuss some general considerations

to inform future work. Simulations were performed for water vapor using a diffusivity of 2.88 x $10^{-5}$ ($m^2 s^{-1}$) and varying

the mass accommodation coefficient from 0.1 to 1.

The sample flow in the aircraft inlet section is mainly impacted by the upstream flow conditions. Thus, the gas sampling

efficiency of the aircraft inlet section can be normalized for both altitudes and both species into a single fit function by using

the volume flow rate divided by the laminar diffusivity as a new variable (Fig. S4). As such, the gas-transmission efficiency





for this section of the inlet system can be parameterized and solved analytically for different accommodation coefficient values

for different species. However, no similar unified correlation of gas sampling efficiency can be found for the sampling line

between the two species. This is because gas loss is a result of a complex combination of flow velocity, flow turbulence, and

laminar diffusivity. In contrast, the laminar diffusivity plays a smaller role in the aircraft inlet section (Fig. 6a), owing to the

larger role of turbulence in this section of the aircraft inlet system.

We have conducted a series of sensitivity studies to illustrate the transmission for a straight tube. Figure 8 presents the results

of gas sampling efficiency in a 33" (0.84 m) straight tube between the two species with different mass accommodation

coefficients at both ground level and high altitude for future interpolations. The laminar diffusivity of a species plays a

significant role in the prediction of the gas transmission efficiency when the mass accommodation is close to 1. However, this

dependence on laminar diffusivity becomes negligible when the mass accommodation is less than 0.25. This paper emphasizes

the importance of choosing the correct diffusivity and mass accommodation coefficient in predicting gas sampling efficiency

of condensable vapors, in particular for highly diffusive condensable vapors (e.g., $NH_3$). Additionally, Khalizov et al. ( 2020)

estimate that the limits of detection of oxidized mercury species are probably similar to those reported for $H_2SO_4$ (Eisele and

Tanner, 1993; Mauldin et al., 1998; Zheng et al., 2010). Given the complexities of sampling oxidized mercury (Elgiar et al.,

2024) significant transmission losses can be expected, with implications for the attainable detection limits in future attempts

to sample oxidized mercury species using atmospheric pressure CIMS techniques (Khalizov et al., 2020). Moreover, the

chemical properties of the inlet line can further influence transmission (e.g., pH, surface composition, reactivity, etc.).

Determining these parameters and their transmission in flight warrants future work that is beyond the scope of this paper.

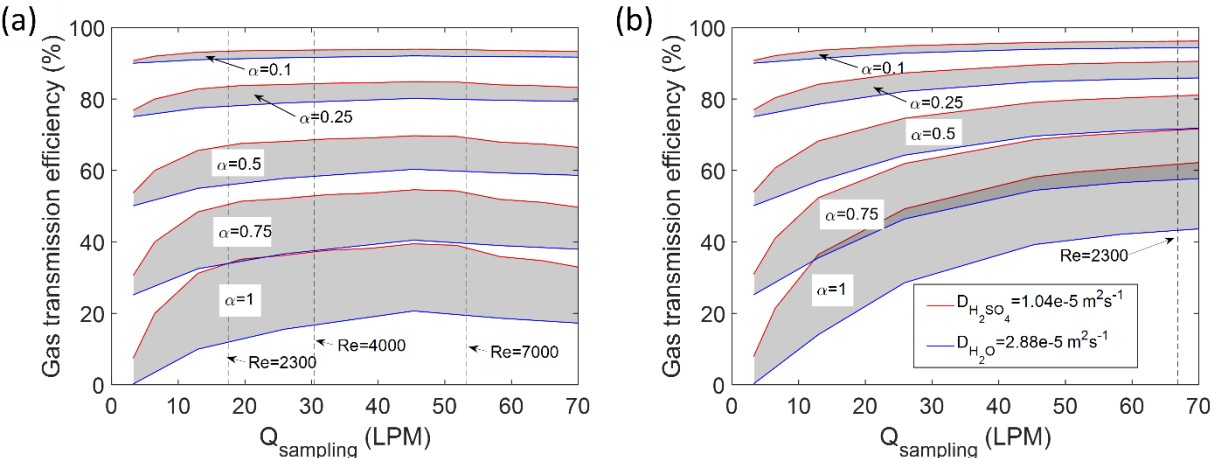

**Figure 8. The comparisons of gas sampling efficiency between water vapor and gas-phase $H_2SO_4$ at the sampling line section. (a)
The correlation of gas sampling efficiency in 33" (0.84m) straight tube at ground level condition (1013 mbar). (b) The correlation of
gas sampling efficiency in 33" (0.84m) straight tube at ground level condition (220 mbar). The red line shows the sampling efficiency
of gas-phase $H_2SO_4$ with diffusivity set as 1.04 x $10^{-5}$ $m^2\ s^{-1}$. The blue line shows the sampling efficiency of water vapor with
diffusivity set as 2.88 x $10^{-5}$ $m^2\ s^{-1}$. The shade areas covered the predicted sampling efficiency between water vapor and gas-phase
$H_2SO_4$ at a selected mass accommodation coefficient.**



## 4. Summary

In this paper, we describe an experimental approach to measure gas transmission efficiency of an aircraft inlet system using $H_2SO_4$ vapor and a high-speed wind tunnel setup. We use these observations to evaluate a CFD model of the aircraft inlet system; and use the validated CFD simulations to predict the overall sampling efficiency of gas-phase $H_2SO_4$ over a range of in-flight sampling conditions that span from the marine boundary layer (MBL) into the upper troposphere and lower stratosphere (UTLS). The inlet transmission for a condensable vapor like $H_2SO_4$ can be optimized using the sampling flow

rate as a variable. For a realistic range of sampling flow rates, increasing the sampling flow rate can help cut transmission losses, and double the overall sampling efficiency of gas-phase $H_2SO_4$ compared to laminar flow conditions. The transmission of $H_2SO_4$ can reach up to 40% in the UTLS, and up to 20% in the MBL, owing primarily to the lower turbulent intensity and shorter residence time of air inside the sampling line under conditions typical of the UTLS.

Experiments and simulation results consistently support the conclusion that increasing the sampling flow rate can increase the gas transmission efficiency. These results challenge the widely accepted assumption that laminar core sampling, i.e., sampling from the core flow inside of an inlet line operated under laminar flow conditions, minimizes wall losses. This assumption is currently very widely used in laboratory and field experiments that sample ambient trace gases and aerosol species. Instead, shortening the residence time that air spends inside inlet lines is found to minimize wall losses well into the turbulent flow

regime. The maximum inlet transmission is found for Re ~ 6000, with lower transmission at both lower and higher Re numbers. For a condensable vapor like $H_2SO_4$, the effects to optimize transmission losses by optimizing the residence time can be as large as a factor of 2 over laminar core sampling.

   To predict the transmission efficiency for other gas-phase species through this aircraft sampling system, the appropriate

selection of the gas-phase diffusivity ($D_{ij}$) and mass accommodation coefficient ($\alpha_i$) of the species (i) is necessary. The overall sampling efficiency is calculated by combining the transmission efficiencies of the aircraft inlet and sampling line. For the aircraft inlet, a simulation-based function predicts transmission efficiency based on $D_{ij}$ and $\alpha_i$, as the gas loss is affected only by the upstream conditions regardless of altitude. Other factors can further affect transmission, i.e., for reactive species multiphase chemistry in addition to physical factors need to be considered. Due to the complexity and feasible computational

cost of modeling gas loss in the straight sampling line, it is recommended to directly calculate the line's transmission efficiency using simulation models. Our flow analysis illustrates that the exact value of $D_{ij}$ only plays a role for sticky gases with $\alpha_i >$ 0.25. We provide simulation data for two species (gas-phase $H_2SO_4$ and water vapor) under various tube lengths and sampling conditions for future interpolation in sampling lines. By combining the transmission efficiencies from all sections and incorporating the sampling line's shape factor, the overall sampling efficiency of a gas-phase species for this aircraft-based

sampling system can be estimated.



**Author contribution:**

RV designed research. DY developed CFD model and conducted the simulations. RV, RM and EA carried out the wind tunnel experiments. DY analysed the data. DY, SD and RV wrote the manuscript, with contributions from all coauthors.

**Competing interests:**

At least one of the (co-)authors is a member of the editorial board of Atmospheric Measurement Techniques.

**Acknowledgment**

Financial support from US National Science Foundation awards AGS-2023961 and AGS-2027252 is gratefully acknowledged.
Wind tunnel testing was conducted at the US Air Force Academy Aeronautics Research Center under Commercial Test Agreement 21-161-AFA-01.

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
