# Peer review of "Laminar gas inlet – Part 2: Wind tunnel chemical transmission measurement and modelling"

_EGUsphere, 2024_

## Author Comment (AC1)

This is a short review as all aspects of the paper were not evaluated: it is too long, thus the Fair rating on Presentation. I gave it Fair on Scientific quality for reasons detailed below. I gave it Good for Significance as the loss of sticky species on inlets is important to get correct. I think with a rewrite and a careful paring down of the text along with addressing the science quality issues below, it could be a pretty good report.

Thank you for reviewing our paper and for the insightful questions. Our manuscript addresses the referee's comments as follows. We have gone through the text to pair it down as best possible. The previous sections 2.4 and 3.2 have been moved to the supplement information section. We believe that the content provided in the revised version is essential for supporting our study. It also strikes a balance to provide the reader with sufficient context about the discussions in part 1 of this paper series, without repeating information that is described in details elsewhere: https://doi.org/10.5194/amt-17-1463-2024

1) Wall loss is a tricky thing to model. How is it incorporated in Fluent? OF course no loss (set species to have no flux at the surface) and diffusion limited loss (set species to have zero concentration at the surface) are conceptually easy to understand and to implement in Fluent. How does one address mass acc. coefficients other than 0 and 1 in Fluent?

As noted in Section 2.3, lines 190–201, the mass accommodation coefficient is defined as the ratio of gas molecules taken up by the surface to the total number of gas-surface collisions. In the revised paper, we have added the following text:

" To simulate different mass accommodation coefficients, we set different species mass fraction boundary condition on the inlet walls. Using this approach, the mass accommodation coefficient is effectively set between the limits of 0 and 1, depending on the species mass fraction on the walls. The influence of varying wall mass fraction boundary conditions on the species accommodation is discussed in our previous paper (Yang et al., 2024)."

2) Related to that issue, the mass acc. results do not make sense in the laminar realm. At 298 K and using 98 for molar mass: (i) At 1 atm (or 0.85 atm? as in Colorado Springs), the diffusion limited loss rate in a 1cm ID tube is about 1.5 s-1. (ii) The kinetic limit (for a mass acc. = 1) is about 2.3x10^4 s-1. What this means is that mass acc. values greater than about 0.001 are at the diffusion limit and there should be very little to no dependence on mass acc. for the throughput of the sampling tube.

While the reviewer is correct that these two rates represent one set of extremes in gas loss rates, we do not agree that these limits represent the role of accommodation coefficient on gas loss. Both the values calculated by the reviewer represent the case of unit

accommodation coefficient, applicable for very different spatial ranges (the kinetic theory for gas loss calculation is applicable if the tube ID is comparable to the mean free path – which is not the case for the 1cm tube considered here).

As noted, at 1 atm, the diffusion limited loss rate for a 1 cm ID tube is ~ 0.1 s-1 (our back of the envelope calculation is off by a factor of 10 from the reviewer's), which is about 10^-5 of the kinetic limit. Whatever the exact value of the diffusion loss rate is, it is representative for loss rate with an accommodation coefficient of 1. A lower accommodation coefficient will lower the diffusion loss from this estimate. To model the accommodation coefficient, we believe that two possible approaches could be used: proportionally scale the diffusion coefficient or tailor the boundary condition as we have done here.

3) Another related issue is the diffusion coefficient. It looks like the authors are using the pressure independent value for all pressures: it actually has units of atm.cm2/s. Diffusivity at altitude will be some 6 or 7 times that at sea level due to the pressure change. Yet it is colder so apply a typical T^1.75 factor. What temperature is the sampling tube at altitude?

Yes, the laminar diffusivity, $D_{ij}$, which describes the diffusivity between the studied species in dry air, is set as a constant. The turbulent diffusivity, however, varies with local flow conditions in the model. In turbulent flow, the contribution of laminar diffusivity to the overall gas-phase diffusion is much (orders of magnitude) smaller than the turbulent diffusivity. For comparison, the combined effect of lower pressure and temperature increases the laminar diffusivity at by about factor ~4 (an upper limit for most conditions during tropospheric sampling).

The original manuscript had already recognized that diffusivity significantly impacts the prediction of gas transmission efficiency, particularly within the transport tube (see Section 3.5, and the original Fig. 8, now Fig. 7). In the revised manuscript, we have added the following sentence to the discussion of Figure 7.

"The shaded area shown in Figure 7 (factor ~3 higher laminar diffusivity) also approximately illustrates the magnitude of the combined effect that higher laminar diffusivity at lower pressure and temperature has on the gas transmission efficiency when sampling at altitude. For most tropospheric sampling, this effect will be minor for non-sticky molecules, but it gains in relative importance for sticky molecules sampled at low pressures; which was not the focus of the current study (windtunnel experiments at ground conditions), and deserves further attention in the future for high altitude sampling."

The temperature along the sampling tube is not really well characterized. Only the ram heating effect on turbulent diffusivity is captured in our calculations. We have clarified this in the revised manuscript with the following sentences:

"The further warming of the sample in the tube as air transfers  into the instrument aboard the aircraft depends on many parameters, incl. temperature gradients towards the cabin air, flow rates, heat transfer from the tube to the gas, etc. and this is currently not well characterized and cannot be generalized; this is not easily possible to model either. We are planning to measure this gradient during an upcoming campaign to further characterize the flow conditions inside the sampling tube in-flight. "

---

## Author Comment (AC2)

**General Comments**

Overall a good paper but some sections could be decreased as there is a long lead up to the main results on the gas transmission efficiency of the inlet. Saying that, I would encourage the authors to expand the methods section to include more information on the operation and setup of the CIMS as these measurements are fundamental to the paper. Furthermore, it is not clear to me how widely applicable the results are to other aircrafts/instruments based on the experimental conditions and assumptions used throughout the study. For example, some CIMS instruments would sub-sample a smaller flow from the sample line or some sample lines may experience a temperature gradient due to differences between the ambient and cabin temperature. It would be useful to clarify the broader applicability of the findings.

Thank you very much for the positive feedback and suggestions. We are addressing your comments in the following sections.

We have moved the previous sections 2.4 and 3.2 into the Supplementary Information, and added a paragraph to describe the operation and setup of the CIMS instrument.

The results from this paper that are transferrable is that for long sample lines laminar core sampling is not recommended for short-lived or reactive species; rather the sampling efficiency increases as the flow rate is increased into the turbulent sampling flow regime. This is established for the first time to the best of our knowledge. The paper further lays the early ground-work for the sampling efficiency for other species using this aircraft inlet, with lessons that are transferrable also to other sampling setups from aircraft. However, it is not the objective of this work to deal with all of the challenges of sampling condensable trace gases from research aircraft, or how temperature gradients in other setups than that used in this study affect the loss of gas-phase species during transportation.

**Specific Comments**

1. Introduction paragraph one – some statements are repeated multiple times and disrupts the flow of the paragraph (e.g. importance of condensable vapours for aerosol growth and hence health). This could be rewritten so that it is clearer.

The first paragraph has been edited and shortened in the revised paper.

2. Line 43 – The sentence on the relevance of trace gases currently reads as this is an exhaustive list. Should be made clear that these are examples.

The sentence has been rewritten in the revised paper as follows:

"Furthermore, trace gases are relevant for atmospheric chemistry in a number of ways, including in the formation and depletion of ozone, establishing the atmospheric oxidative capacity, and the oxidation of mercury, a potent neurotoxin (Khalizov et al., 2020; Shah et al., 2021)."

3. Line 45 – Understanding the formation and growth of short-lived reactive gases. Suggest change word growth, this feels more appropriate to describe aerosols.

The reference to growth has been removed from the revised sentence.

4. Line 110 – What is the material of the sampling tube?

The material has been added. Its stainless steel.

5. Line 114 – What is the range of flow rates sampled by the CIMS?

The sampling flow rate of CIMS is listed in Table 1. We have added reference to Table 1 here.

6. Fig1d – What does the dashed line represent?

We have added the following sentence in the caption to Figure 1 of the revised paper:

"The vertical dashed line marks the first section, which is the same length for each setup. For Type 1, the dashed line represents the different shape, but same length as Type 0."

7. Line 135-136 – What concentrations are used for each of the reagents? And what is the resulting concentration of H2SO4?

The concentrations of reactant mixtures, and typical H2SO4 concentrations have been added in the revised paper.

8. Line 174 – Later on in the paper you mention the different humidity conditions in the wind tunnel across the experiment period. Is it correct that H2SO4 is diffusing in **dry** air?

The reference to dry air describes the model treatment of binary diffusion. The revised manuscript clarifies that the model uses the H2SO4 diffusion coefficient in dry air. The later reference to different humidity conditions is in a different context, as humidity is relevant primarily as it affects the production of OH radicals.

9. Line 176 - In addition, as the temperature gradient in the transmission line is insignificant, we neglect thermal diffusion loss. Does this remain true for ambient sampling where there can be larges differences between the cabin and ambient temperatures?

This statement applies to the experimental conditions probed at the windtunnel. The local sampling conditions in-flight are different. The diffusion coefficient increases at lower pressures and colder temperatures. A similar comment was made by reviewer #1. See our detailed response to reviewer #1, comment 3. And the discussion added in the revised manuscript in relation to the discussion of sensitivity studies how the value of the diffusion coefficient changes the gas-sampling efficiency.

10. Line 229 - These ion concentrations were recorded under different operating conditions by CIMS. Different inlet or CIMS operating conditions? If CIMS what are these different conditions and what is the rationale for this?

The 'different operating conditions' in this paragraph refer to the various experimental setups illustrated in Figure 1 and the different sampling flow rates listed in Table 1. We have added these references following this sentence.

11. Line 314 – Can you include a schematic of the NO3 CIMS in the methods that highlights the IMR region you are describing here.

We have added a brief description of the NO3 CIMS instrument, including a schematic of the IMR region in Section 2.2. of the revised paper.

[Figure]

12. Line 315 – I would be explicit here that the lower signal response at 16 SLM is specific to the instrument used in this study and you cannot be certain that this holds true for other CIMS instruments that are operated under different conditions.

We agree, and have made this explicit in the revised manuscript:

"This is specific to the instrument configuration used here, and related to the reaction time and flow characteristics inside the ion molecule reaction chamber of the CIMS instrument"

13. Fig 6 – it would be helpful to the reader to define Q in the caption as this is defined later in the paper.

We changed the symbol for sampling flow rate in later paragraphs from 'Q' to '$Q_{sampling}$'. And remain consistent for all figures.

**Technical Comments**

1. Line 17 – remove using

Typo fixed

2. Line 40 – composition-dependent, . remove comma

The sentence has been rephrased based on the Specific Comments 1

3. Line 56 – replace aboard with onboard

Typo fixed

4. Line 109 – replace aboard with onboard

Typo fixed

5. Line 161 - sampling tube designs use the commercial code – needs rewording

The sentence has been rewritten.

6. Line 174 – replace refer with referred

Typo fixed

7. Line 237 - ($H2SO4\ NCPS$,) – remove comma

Typo fixed

8. Line 338 – Hanson et all., remove et al as Hanson only author

Typo fixed

9. Fig5 caption – description of chapter 2.4. Change to section 2.4

Typo fixed

10. Line 397 - This is due to sample flow is more turbulent. Typo

Typo fixed